# Roles of the Caspase-11 Non-Canonical Inflammasome in Rheumatic Diseases

**DOI:** 10.3390/ijms25042091

**Published:** 2024-02-08

**Authors:** Young-Su Yi

**Affiliations:** Department of Life Sciences, Kyonggi University, Suwon 16227, Republic of Korea; ysyi@kgu.ac.kr; Tel.: +82-31-249-9644

**Keywords:** caspase-11, caspase-4, non-canonical inflammasome, rheumatic disease, inflammasome

## Abstract

Inflammasomes are intracellular multiprotein complexes that activate inflammatory signaling pathways. Inflammasomes comprise two major classes: canonical inflammasomes, which were discovered first and are activated in response to a variety of pathogen-associated molecular patterns (PAMPs) and danger-associated molecular patterns (DAMPs), and non-canonical inflammasomes, which were discovered recently and are only activated in response to intracellular lipopolysaccharide (LPS). Although a larger number of studies have successfully demonstrated that canonical inflammasomes, particularly the NLRP3 inflammasome, play roles in various rheumatic diseases, including rheumatoid arthritis (RA), infectious arthritis (IR), gouty arthritis (GA), osteoarthritis (OA), systemic lupus erythematosus (SLE), psoriatic arthritis (PA), ankylosing spondylitis (AS), and Sjögren’s syndrome (SjS), the regulatory roles of non-canonical inflammasomes, such as mouse caspase-11 and human caspase-4 non-canonical inflammasomes, in these diseases are still largely unknown. Interestingly, an increasing number of studies have reported possible roles for non-canonical inflammasomes in the pathogenesis of various mouse models of rheumatic disease. This review comprehensively summarizes and discusses recent emerging studies demonstrating the regulatory roles of non-canonical inflammasomes, particularly focusing on the caspase-11 non-canonical inflammasome, in the pathogenesis and progression of various types of rheumatic diseases and provides new insights into strategies for developing potential therapeutics to prevent and treat rheumatic diseases as well as associated diseases by targeting non-canonical inflammasomes.

## 1. Introduction

Inflammation is an innate immune response that protects the body from pathogenic infections and cellular damages [1]. An inflammatory response occurs when inflammatory cells sense the presence of pathogen-associated molecular patterns (PAMPs) or damage-associated molecular patterns (DAMPs) through their pattern-recognition receptors (PRRs) [1]. The response consists of two successive steps: priming and triggering [2,3]. The priming step prepares for an inflammatory response by increasing the expression of inflammatory molecules, whereas the triggering step activates and boosts this response by activating inflammasomes and cytosolic protein complexes, which act as molecular platforms to activate inflammatory signaling pathways [4,5]. The nucleotide oligomerization domain-like receptor (NLR) family of inflammasomes, absent in melanoma 2 (AIM2), and pyrin inflammasomes that were previously discovered are canonical inflammasomes, whereas the recently identified mouse caspase-11 and human caspase-4/5 inflammasomes are non-canonical inflammasomes, signaling platforms that are activated through sensing intracellular lipopolysaccharide (LPS) [6,7,8,9,10,11]. Inflammasome activation triggers two major inflammatory responses: the induction of gasdermin D (GSDMD) pore-mediated inflammatory cell death, known as pyroptosis, and the caspase-1-promoted secretion of inflammatory cytokines through GSDMD pores, leading to the augmentation of inflammatory responses [6,12,13]. Numerous studies have demonstrated that inflammasomes are key players in inflammatory responses and various human pathological conditions, providing evidence that they could be pharmacological targets for such conditions [14,15,16,17,18,19]. However, these studies mainly focused on canonical inflammasomes. Interestingly, accumulating evidence indicates that non-canonical inflammasomes, especially caspase-11 non-canonical inflammasome, play regulatory roles in inflammatory responses and human diseases [20,21,22,23,24,25,26,27,28,29,30,31], suggesting that they could also be the basis for developing potential therapeutics.

Rheumatic diseases are a group of chronic inflammatory and autoimmune diseases that primarily affect the joints, muscles, and connective tissues and can result in substantial morbidity. A large number of patients worldwide suffer from over 200 rheumatic diseases [32]. These include rheumatoid arthritis (RA), osteoarthritis (OA), gouty arthritis (GA), systemic lupus erythematosus (SLE), Sjögren’s syndrome (SjS), psoriasis, and ankylosing spondylitis (AS). Although the etiology of rheumatic diseases is uncertain, inflammatory and autoimmune responses are considered major risk factors for disease onset and progression [33,34,35]. Interestingly, recent studies have demonstrated that non-canonical inflammasomes play roles in various rheumatic diseases. This review summarizes such research on the regulatory roles of non-canonical inflammasomes, particularly the caspase-11 non-canonical inflammasome, in the pathogenesis of rheumatic diseases and discusses scientific evidence for the development of antirheumatic therapeutics that target non-canonical inflammasomes.

## 2. Caspase-11 Non-Canonical Inflammasome

### 2.1. Structure of Caspase-11 Non-Canonical Inflammasome

The first non-canonical inflammasome was discovered in the 129S6 mouse strain and expresses a truncated and non-functional caspase-11 protein [8]. Unlike other caspases, which mediate apoptosis, caspase-11 was discovered as an inflammatory caspase that directly recognizes intracellular lipopolysaccharide (LPS) and induces inflammatory responses by forming a caspase-11 non-canonical inflammasome [8,9,10]. Although caspase-11 is functionally distinguished from apoptotic caspases, its molecular structure is well conserved among caspase family members and is very similar to the latter. Caspase-11 comprises three domains: a caspase recruitment domain (CARD) at the amino terminus, followed by two catalytic domains, a larger catalytic domain, p20, and a smaller catalytic domain, p10, at the carboxyl-terminus (Figure 1A). Since the discovery of mouse caspase-11, efforts have been made to identify human counterparts of mouse caspase-11. A biochemical study revealed that caspase-4 and caspase-5 are human inflammatory caspases equivalent to mouse caspase-11 [11], and recent studies have further confirmed that human caspase-4/5 are homologs of mouse caspase-11 [31,36,37,38]. Similar to mouse caspase-11, human caspase-4/5 also comprises three domains: an amino-terminal CARD, p20, and a carboxyl-terminal p10 (Figure 1A); however, the sizes of these caspases are different. Mouse caspase-11, human caspase-4, and caspase-5 are 373, 377, and 434 amino acids in length, respectively (Figure 1A).

### 2.2. Caspase-11 Non-Canonical Inflammasome-Activated Inflammatory Signaling Pathways

Inflammasomes are activated in response to specific stimuli, and a broad range of PAMPs and DAMPs have been identified as canonical inflammasome stimuli [5,39,40]. After the discovery of the caspase-11 non-canonical inflammasome, subsequent studies successfully demonstrated that it is activated by direct sensing of cytosolic lipopolysaccharide (LPS), the most conserved endotoxin, and PAMP in the cell walls of Gram-negative bacteria [8,9,10]. LPS is originally derived from the extracellular infection by the Gram-negative bacteria and then enters the cells via Toll-like receptor 4 (TLR4)-mediated endocytosis, the cell surface receptor for advanced glycation end-product (RAGE)-mediated endocytosis, or bacterial outer membrane vesicle (OMV)-mediated internalization [3]. Cytosolic LPS is the only stimulus that activates the caspase-11 non-canonical inflammasome. Caspase-11 directly senses cytosolic LPS through an interaction between the CARD of caspase-11 and lipid A of LPS (Figure 1B). The direct sensing of LPS by caspase-11 induces the oligomerization of LPS-caspase-11 complexes through direct CARD-CARD interactions. It is then activated through the autoproteolysis of caspase-11 at the 285 aspartic acid residue by the 254 cysteine residue, which promotes enzymatic activity for caspase-11 autoproteolysis, resulting in the generation of active caspase-11 p26 fragments (Figure 1B) [41].

Caspase-11 p26 generated by the activation of the caspase-11 non-canonical inflammasome triggers two main inflammatory signaling pathways: GSDMD pore-mediated pyroptosis and the secretion of pro-inflammatory cytokines, IL-1β and IL-18, from the cells. The active caspase-11 p26 triggers the proteolytic processing of GSDMD at aspartic acid 276, generating both amino-terminal GSDMD (N-GSDMD) and carboxyl-terminal GSDMD (C-GSDMD) fragments. N-GSDMD fragments move to cell membranes and generate N-GSDMD pores with 31- to 34-fold symmetry, leading to N-GSDMD pore-induced rupture and inflammatory cell death, known as pyroptosis [2,42]. The active caspase-11 p26 also triggers the maturation and secretion of IL-1β and IL-18, through functional cooperation with the NLRP3 canonical inflammasome. The caspase-11 non-canonical inflammasome can activate the NLRP3 inflammasome by promoting potassium (K^+^) efflux, a key event required for the activation of the NLRP3 canonical inflammasome, through pyroptosis-mediated cell membrane damages and membrane gate proteins, including the P_2_X7 channel, bacterial pore-forming toxins, and the pannexin 1 channel [43,44,45]. Activation of the NLRP3 inflammasome leads to the NLRP3 inflammasome-mediated proteolytic activation of caspase-1. The proteolytically activated caspase-1 in turn promotes the proteolytic maturation of inactive pro-IL-1β and pro-IL-18 to generate active IL-1β and IL-18, and these are secreted from the inflammatory cells through the N-GSDMD pores [2,42]. These secreted pro-inflammatory cytokines augment additional inflammatory responses by activating other types of immune cells. Thus, caspase-11 non-canonical and NLRP3 canonical inflammasomes functionally cooperate in inflammasome-activated inflammatory signaling pathways. Recent mechanistic studies have demonstrated that the caspase-11 non-canonical inflammasome activates the NLRP3 inflammasome by inducing potassium ion (K^+^) efflux, a key determinant of NLRP3 inflammasome activation, through pyroptosis-mediated membrane damage, bacterial pore-generating toxins, and membrane gate proteins, such as the pannexin I channel and the P_2_X7 receptor [3,46]. This observation strongly suggests that canonical and non-canonical inflammasomes mediate cooperative rather than independent inflammasome-activated inflammatory signaling pathways. The caspase-11 non-canonical inflammasome-activated inflammatory signaling pathways and their functional cooperation with the NLRP3 inflammasome are shown in Figure 1B.

## 3. The Roles of Caspase-11 Non-Canonical Inflammasome in Rheumatic Diseases

### 3.1. Inflammatory Arthritis—Rheumatoid Arthritis (RA) and Gouty Arthritis (GA)

Arthritis is defined as the inflammation or swelling of one or more joints and describes more than 100 conditions that affect the joints, tissues around the joint, and other connective tissues. Specific symptoms vary depending on the type of arthritis, but usually include joint pain and stiffness. Arthritis can be broadly classified into two categories: inflammatory and non-inflammatory. Inflammatory arthritis is a category of arthritis caused by severe inflammation and characterized by swelling, pain, warmth, and joint tenderness, whereas non-inflammatory arthritis is often the result of a breakdown or damage to the different parts of the joints, such as the cartilage. As most cases of inflammatory arthritis are systemic, symptoms associated with inflammatory arthritis can occur in other parts of the body, such as eye inflammation, skin rashes, dry mouth, hair loss, and fever. There are many types of inflammatory arthritis, and its most prevalent form is rheumatoid arthritis (RA). RA is a type of arthritis where the immune system attacks the tissue lining the joints on both sides of the body. In some individuals, inflammation associated with RA can damage a wide variety of body systems, including the skin, eyes, lungs, heart, and blood vessels [47]. RA occurs when the immune system mistakenly attacks the tissues of the body and affects the lining of the joints, causing painful swelling that can eventually result in bone erosion and joint deformity [47]. The prevalence and disease burden of RA differ between geographic regions, with a high prevalence in industrialized countries. However, the global prevalence of RA has been approximately 0.5% over the past three decades [48].

Underscoring the association between inflammation and pathogenesis, several previous studies have demonstrated that inflammasomes play roles in RA; however, these studies focused mainly on canonical inflammasomes, particularly the NLRP3 inflammasome [49,50,51,52]. Interestingly, recent studies have reported that non-canonical inflammasomes also play a role in RA. The recruitment and accumulation of inflammatory cells such as neutrophils and macrophages are a key step in the onset and progression of various inflammatory diseases, including RA [53]. Cox et al., investigated the role of the caspase-11 non-canonical inflammasome in association with matrix metalloproteinase-8 (MMP-8) in neutrophil migration in inflammatory arthritis. Increased neutrophil infiltration in synovial tissues and more severe symptoms have been observed in MMP-8-deficient mouse models of inflammatory arthritis induced by Freund’s complete adjuvant (FCA) [54]. Genomic analysis revealed a reduction in caspase-11 gene expression in the neutrophils of MMP-8-deficient inflammatory arthritic mice [54], suggesting that the caspase-11 non-canonical inflammasome inhibits neutrophil migration to synovial tissues and induces neutrophil apoptosis in the tissues, thereby alleviating arthritis symptoms through functional cooperation with MMP-8. It is very interesting that, rather than a general role, the caspase-11 non-canonical inflammasome showed an anti-inflammatory activity in the MMP-8-deficient neutrophils in this study. The role of the caspase-11 non-canonical inflammasome might be different depending on the animal models and cell types, and further studies in this regard will be required. Also, despite the evidence from this study that the caspase-11 non-canonical inflammasome and MMP-8 are involved in inflammatory arthritis, functional relationships and cooperation between these two molecules need to be investigated further. As mentioned above, caspase-4 is the human homologue of mouse caspase-11 [11]. RA synovial fibroblasts (RASFs) are effector cells that contribute to synovial hyperplasia or pannus formation, resulting in joint inflammation and bone erosion [55]. Yang et al. demonstrated the role of the human caspase-4 non-canonical inflammasome in the apoptosis of RASFs, which contributes to arthritic cartilage degradation. Blockade of the caspase-4 non-canonical inflammasome inhibits RASF apoptosis by attenuating vimentin cleavage and p53 nuclear translocation [56]. Since vimentin was shown to induce resistance to apoptosis by inhibiting p53 and since vimentin cleavage promotes apoptosis [57,58], these results indicate that the caspase-4 non-canonical inflammasome induces the apoptosis of RASFs by modulating vimentin cleavage and p53 nuclear translocation. This further suggests that the caspase-4 non-canonical inflammasome may play a unique role in attenuating RA by increasing the sensitivity of RASFs to apoptosis. However, the above-mentioned evidence was obtained through in vitro experiments using RASFs; therefore, the role of the caspase-4 non-canonical inflammasome in RA should be further confirmed in animal models of inflammatory arthritis and clinically in RA patients.

Notably, these studies demonstrated the regulatory role of the caspase-11 non-canonical inflammasome in inflammatory arthritis under sterile conditions of the inflammatory response. However, bacterial infections in the fluid and tissues of joints can also cause RA, and LPS in Gram-negative bacteria can induce caspase-11 non-canonical inflammasome-activated inflammatory responses [8,9,10,59,60]. Several studies have investigated the regulatory role of the caspase-11 non-canonical inflammasome in the pathogenesis of inflammatory arthritis under non-sterile inflammatory conditions induced by bacterial joint infections. Lacey et al., investigated the role of the caspase-11 non-canonical inflammasome and its effectors in joint arthritis induced by infection with the Gram-negative bacterium *Brucella melitensis*. *Brucella* joint infection induces caspase-11 non-canonical inflammasome activation and pro-inflammatory cytokine production, leading to joint inflammation in mice [61]. Additionally, caspase-11 non-canonical inflammasome-mediated pyroptotic cell death in response to *Brucella* infection was promoted in bone-marrow-derived macrophages (BMDMs) [61]. Interestingly, *Brucella* infection activates the caspase-11 non-canonical inflammasome but not the NLRP3 canonical inflammasome in BMDMs [61], suggesting that the caspase-11 non-canonical inflammasome, rather than the NLRP3 inflammasome, may be a more critical player in inflammatory arthritis induced by *Brucella* joint infection in mice. This study demonstrates that Gram-negative bacterial infection in the joints induces joint arthritis by activating the caspase-11 non-canonical inflammasome through pro-inflammatory cytokine production and the pyroptosis of macrophages in mice. Kang et al. also reported the functional cooperation of the caspase-11 non-canonical inflammasome with Prdx1 in the mouse model of inflammatory arthritis caused by joint infection with other types of Gram-negative bacteria such as *Listeria monocytogenes* and *Escherichia coli*. The level of circulating Prdx1 is increased in the joint tissues of the arthritic mice infected with bacteria, and the blockade of Prdx1 ameliorates arthritic symptoms in the joint tissues of the mice infected with bacteria [62]. In addition, Prdx1 release under arthritic conditions was decreased in the caspase-11 KO mice [62], further suggesting a functional relationship between the two molecules in the pathogenesis of joint arthritis during Gram-negative bacterial infection in mice. However, the molecular interactions and underlying mechanisms need to be investigated in further detail.

Gouty arthritis (GA) is another common type of inflammatory arthritis. It is caused by sharp crystals of uric acid that form in and around the joints and is characterized by pain, swelling, heat, redness, and tenderness in the joints, especially in the foot, ankle, or knee [63]. The prevalence and incidence of GA vary worldwide with a prevalence of less than 1% to 6.8% and an incidence of 0.58–2.89 per 1000 person every year [63]. While previous studies have demonstrated that canonical inflammasomes, particularly the NLRP3 inflammasome, are important factors in inducing inflammatory responses and leading to the exacerbation of GA [64,65,66,67], the involvement of non-canonical inflammasomes in the pathogenesis of GA has not drawn much attention. However, Caution et al. recently investigated the functional cooperation of the caspase-11 non-canonical inflammasome with cofilin in neutrophil chemotaxis and extracellular trap formation during acute GA in a monosodium urate (MSU)-induced mouse model of GA. The absence of the caspase-11 non-canonical inflammasome reduces the influx of macrophages and neutrophils into disease lesions and the production of neutrophil extracellular traps (NETs) in the disease mice [68]. It also decreases the production of gout-specific pro-inflammatory cytokines, leading to the amelioration of the disease in mice [68]. Interestingly, a lower production of NETs in the caspase-11-deficient neutrophils was associated with altered cofilin phosphorylation in response to MSU in GA mice [68]. This study demonstrates that the caspase-11 non-canonical inflammasome exacerbates GA by promoting the trafficking and functions of neutrophils and macrophages, and by producing GA-specific pro-inflammatory cytokines in cooperation with cofilin in an acute mouse model of GA. However, the detailed mechanisms underlying the functional relationship between the caspase-11 non-canonical inflammasome and cofilin in GA pathogenesis require further investigation.

Taken together, these studies indicate that mouse caspase-11 and human caspase-4 non-canonical inflammasomes are activated and regulate the pathogenesis of inflammatory arthritis and RA (Figure 2) as well as GA (Figure 3). Interestingly, the roles of these non-canonical inflammasomes vary depending on the disease type, and are even different in the same disease depending on the inflammatory conditions and animal models. Moreover, these non-canonical inflammasomes functionally cooperate with other cellular molecules, such as MMP, Prdx1, and cofilin, during the pathogenesis of RA and GA. These findings emphasize the need for further mechanistic studies of the caspase-11 non-canonical inflammasome-triggered pathogenesis of these diseases and the identification and validation of novel regulators associated with caspase-11 non-canonical inflammasomes.

### 3.2. Osteoarthritis (OA)

Osteoarthritis (OA) is a type of degenerative joint disease that results from the breakdown of joint cartilage and the underlying bone. OA most commonly affects the joints in the hands, knees, and hips and is characterized by pain, stiffness, tenderness, swelling, loss of flexibility, and bone spurs. OA affects approximately 7% of the global population, including 10% of men and 18% of women age 60 and older [69].

Previous studies have demonstrated that canonical inflammasomes and the NLRP3 inflammasome are actively involved in the pathogenesis of OA and that selective targeting of canonical inflammasomes is an emerging pharmacological strategy for OA therapy [70,71,72,73,74]. Accumulating evidence has revealed that non-canonical inflammasomes are possible players in OA pathogenesis. Apoptosis of chondrocytes is a major risk factor for the initiation and development of OA. Qin et al., investigated the role of the human caspase-4 non-canonical inflammasome and the potential mechanism involved in LPS-induced chondrocyte apoptosis [75]. LPS induces apoptosis in human chondrocytes isolated from patients with OA, and the caspase-4 non-canonical inflammasome is activated in the LPS-stimulated human chondrocytes of OA patients [75]. Melatonin, a pineal gland hormone, attenuates the apoptosis of human chondrocytes and inhibits the activation of the caspase-4 non-canonical inflammasome in LPS-stimulated human chondrocytes [75], suggesting the possibility that the activation of the caspase-4 non-canonical inflammasome may contribute to the LPS-induced apoptosis of chondrocytes and the development of OA, which needs to be demonstrated in the future.

Extracellular vesicles (EVs) are lipid-bilayer-delimited particles naturally secreted from cells into the extracellular space [76], and accumulating evidence has demonstrated a functional crosstalk between EVs and inflammasomes in inflammatory responses and diseases [77,78,79,80]. Recently, Ebata et al. investigated the pathogenic role of EVs from inflammatory macrophage and caspase-11 non-canonical inflammasomes in cartilage degradation and OA pathogenesis in murine models of OA. EVs derived from LPS-stimulated macrophages induced the apoptosis of mouse chondrocytes and upregulated expression of chondrocyte catabolic- and pyroptosis-associated factors in mouse chondrocytes [77]. Interestingly, the disruption of caspase-11 alleviates pyroptosis and catabolic processes in macrophage EV-stimulated chondrocytes, resulting in the amelioration of pathogenic changes in collagenase-induced and anterior cruciate ligament transection (ACLT) mouse models of OA [77].

With caspase-4 being the human ortholog of mouse caspase-11 [11], the two studies discussed above provide evidence that human caspase-4 and mouse caspase-11 non-canonical inflammasomes play regulatory roles in OA pathogenesis by promoting chondrocyte pyroptosis [75,77]. Selective targeting of these non-canonical inflammasomes might be a potential strategy to treat OA or to retard its progression. Despite such clear evidence, the molecular and cellular mechanisms by which non-canonical inflammasomes are involved in OA pathogenesis remain to be identified, and translational studies in OA patients using selective caspase-4/11 inhibitors are yet to be conducted. Taken together, these studies show that non-canonical inflammasomes are actively involved in OA pathogenesis by inducing cartilage degradation through pyroptosis and catabolic processes in chondrocytes (Figure 4).

### 3.3. Systemic Lupus Erythematosus (SLE)

Systemic lupus erythematosus (SLE), also known as lupus, is a chronic rheumatic disease that causes systemic inflammation affecting multiple organs, such as the skin, joints, heart, lungs, kidneys, circulating blood cells, and brain and, in some cases, causing permanent tissue damage. The symptoms of SLE include severe fatigue, joint pain and swelling, headache, hair loss, and rashes across the nose and cheeks, known as “butterfly rashes”. The global incidence of SLE is around 5 per 100,000 person-years, and the newly diagnosed population is estimated to be 0.4 million people annually in the world [81]. The global prevalence of SLE is approximately 3.4 million, and approximately 90% of patients are women in the age group of 15–44 [81]. However, the epidemiology of SLE varies considerably between sexes, ages, and geographical regions [81].

Canonical inflammasomes are involved in the pathogenesis of SLE and lupus-related diseases such as lupus nephritis and are regarded as potential targets for SLE treatment [82,83,84,85,86]. Recent studies have also reported the regulatory role of non-canonical inflammasomes in SLE pathogenesis. Kumpunya et al., demonstrated the functional cooperation of the caspase-11 non-canonical inflammasome with cyclic GMP AMP synthase (cGAS) during SLE pathogenesis in a murine lupus model. cGAS deficiency enhances lupus-like symptoms and inflammatory pathology in a pristane-induced mouse model of lupus [87]. cGAS deficiency also increases the expression of the caspase-11 gene, leading to activation of the caspase-11 non-canonical inflammasome in the lungs of pristane-induced lupus mice [87]. These results suggest that the caspase-11 non-canonical inflammasome is activated during SLE pathology by increasing caspase-11 expression, which is inhibited by cGAS.

Extracellular DNA released from cells, such as nuclear and mitochondrial DNA (mtDNA), is known to induce an inflammatory pathology in SLE by forming immune complexes with autoantibodies [88]. Proteolytic activation of GSDMD and GSDMD pore formation, induced by the activation of the caspase-11 non-canonical inflammasome, promotes GSDMD pore-mediated pyroptosis [6,12,13]. Miao et al. demonstrated that mtDNA released from cells induces the activation of GSDMD and caspase-11 non-canonical inflammasomes in SLE patients and murine lupus models. GSDMD was proteolytically activated in the neutrophils of SLE patients as well as in pristane-induced and MRL/lpr mouse models of lupus, and GSDMD-mediated pyroptotic death of neutrophils was increased in the kidneys of these lupus mouse models [89]. In addition, the caspase-11 non-canonical inflammasome is activated in the neutrophils of lupus mice, leading to the proteolytic activation of GSDMD and extracellular DNA release through GSDMD pores [89]. Interestingly, oxidized mtDNA promotes GSDMD oligomerization and pore formation, resulting in the pyroptotic death of neutrophils in mouse models of lupus. Furthermore, GSDMD inhibition in neutrophils alleviates disease severity in mouse models of lupus [89]. These results suggest that the caspase-11 non-canonical inflammasome is activated, leading to the proteolytic activation of GSMD, mtDNA-mediated GSDMD pore formation, and the GSDMD pore-mediated pyroptotic death of neutrophils during SLE pathogenesis. This study clearly showed that activation of the caspase-11 non-canonical inflammasome and GSDMD is a key event in SLE pathology in two lupus mouse models, and that selective inhibition of these two molecules is a possible therapeutic approach to treat SLE. However, since caspase-4 is the human counterpart of mouse caspase-11 [11], the role of the caspase-4 non-canonical inflammasome and GSDMD in human patients with SLE should be further evaluated. Taken together, these two studies suggest that the caspase-11 non-canonical inflammasome plays a role in SLE pathogenesis through functional crosstalk with cGAS, induction of GSDMD activation, and GSDMD pore-mediated pyroptosis in BMDMs and neutrophils (Figure 5).

### 3.4. Sjögren’s Syndrome (SjS)

Sjögren’s syndrome (SjS) is a chronic autoimmune rheumatic disease that can occur alone or in combination with other autoimmune rheumatic diseases such as RA and SLE. SjS occurs when the immune system damages glands that produce and control moisture in the body. The most common symptoms are chronic unusual dryness of the eyes, mouth, or vagina. Other common symptoms include muscle pain, fatigue, and rashes. Less commonly, SjS can affect the nervous system and internal organs such as the lungs, gastrointestinal tract, and kidneys. Although the incidence and prevalence are quite different depending on geographical origins, sex, study design, and diagnostic criteria, the global incidence and prevalence of SjS is approximately 0.01 to 0.05% and 0.5 to 1.0% of the global population, respectively [90,91].

Similar to other rheumatic diseases, canonical inflammasomes, including the NLRP3 inflammasome, are important players in SjS’s pathogenesis and progression [51,92,93,94]. Some studies have reported new roles for non-canonical inflammasomes as well. The damage caused to exocrine glands by immune cell infiltration that destroys glandular homeostasis is a key event in SjS. Bulosan et al., reported the potential involvement of the caspase-11 non-canonical inflammasome in altered glandular homeostasis in an SjS-prone mouse model [95]. The study showed the upregulation of caspase-11 expression, leading to the activation of caspase-1 and production of IL-18 in the macrophages and submandibular glands of SjS-prone C57BL/6.NOD-Aec1Aec2 mice [95]. It also found increased apoptosis of epithelial cells in the submandibular glands of SjS-prone mice [95]. This study discovered that the caspase-11 non-canonical inflammasome, along with its downstream effector, caspase-1, is responsible for increasing epithelial cell death in the submandibular glands and altering glandular homeostasis, resulting in SjS pathogenesis in the SjS mouse model.

Pannexin-1 (Panx1) has been demonstrated to be a factor that induces inflammation and cell death; therefore, it is regarded as a potential target in inflammation [96,97]. Basova et al. investigated the functional roles of Panx1 and caspase-11 non-canonical inflammasomes in lacrimal gland-inflamed mouse model of SjS. Panx1, pro-inflammatory factors, and caspase-11 were upregulated in the mouse lacrimal glands injected with IL-1α, and the blockade of Panx1 increased the epithelial cell progenitor engraftment into the IL-1α-injured lacrimal glands in mice [98]. Additionally, inhibition of Panx1 or caspase-11 reduces inflammation in the lacrimal glands of *TSP-1* KO mice, a mouse model of aqueous-deficient dry eye, by decreasing the expression of pro-inflammatory factors and lymphocyte infiltration [98]. These results suggest that the Panx1 and caspase-11 non-canonical inflammasomes play regulatory roles in the pathogenesis of SjS by inducing lacrimal inflammation in the mouse model of SjS.

Evidence from the two studies discussed in this section show that the caspase-11 non-canonical inflammasome is activated and is involved in the pathogenesis of SLE (Figure 6). However, these studies demonstrated the potential roles of the caspase-11 non-canonical inflammasome only in mouse SjS models, and these observations should be confirmed by examining the roles of the human caspase-4 non-canonical inflammasome in patients with SjS. In addition, the underlying molecular mechanisms, the key factors involved, and the functional relationship between these factors and caspase-4/11 non-canonical inflammasomes in SjS pathogenesis remain to be investigated.

## 4. Concluding Remarks

This review comprehensively discusses our current understanding of the regulatory roles of non-canonical inflammasomes, particularly the caspase-11 non-canonical inflammasome, in the pathogenesis and progression of rheumatic diseases such as RA, infectious arthritis, GA, OA, SLE, and SjS, as well as some of their underlying mechanisms, as summarized in Table 1. Although non-canonical inflammasomes play a unique role in association with different factors in each rheumatic disease, all the studies discussed in this review show that non-canonical inflammasomes induce inflammatory responses in disease lesions, resulting in the onset and progression of rheumatic diseases, strongly suggesting that non-canonical inflammasomes may be potential therapeutic targets for rheumatic diseases.

However, most studies have focused on the mouse caspase-11 non-canonical inflammasome, and its functions have been mainly evaluated in mouse models of rheumatic diseases. These findings suggest that the regulatory roles of the human caspase-4 non-canonical inflammasome need to be demonstrated in human patients with rheumatic diseases. Although the roles of non-canonical inflammasomes in diseases have been demonstrated, their molecular and cellular mechanisms remain largely unknown. Therefore, future studies need to focus on the identification and validation of cellular factors that are functionally associated with non-canonical inflammasomes during the pathogenesis of rheumatic diseases and their underlying mechanisms. Moreover, the development of selective non-canonical inflammasome-specific inhibitors and translational studies using these inhibitors in human patients with rheumatic diseases are also required.

In conclusion, the mouse caspase-11 (and possibly human caspase-4) non-canonical inflammasome plays regulatory roles in the pathogenesis and progression of various types of rheumatic diseases in functional association with other cellular factors. This occurs through the induction of inflammatory responses in disease lesions through pro-inflammatory cytokine production and pyroptotic death of target cells. Understanding the regulatory roles and underlying mechanisms of the caspase-11 non-canonical inflammasome-mediated pathogenesis of rheumatic diseases may contribute not only to the development of promising therapeutics that target caspase-11 non-canonical inflammasomes in these diseases, but also to clinical and translational studies in human patients suffering from rheumatic diseases.

## Figures and Tables

**Figure 1 ijms-25-02091-f001:**
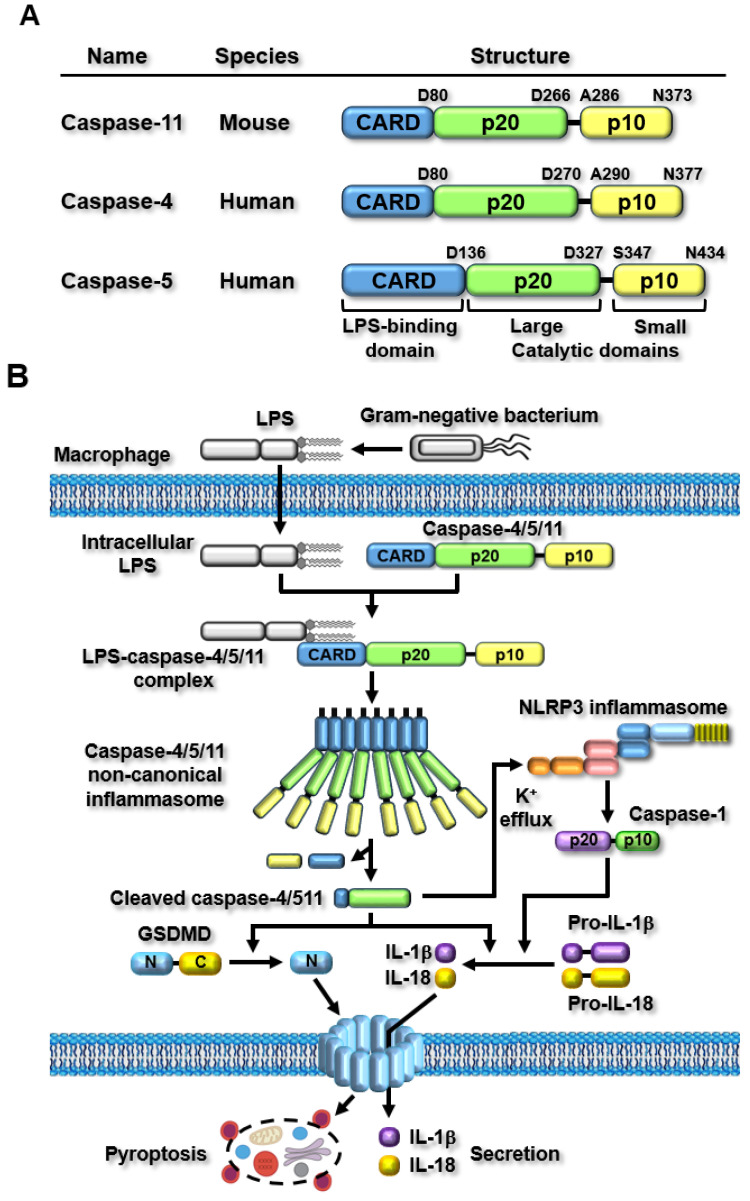
Structure of caspase-11 non-canonical inflammasome and caspase-11 non-canonical inflammasome-activated inflammatory signaling pathways. (**A**) Structural comparison of mouse caspase-11, human caspase-4, and caspase-5. (**B**) LPS originating from Gram-negative bacteria enters the cells, and then caspase-11 senses the intracellular LPS by direct interaction to form LPS-caspase-11 complexes, leading to the generation of caspase-11 non-canonical inflammasome by oligomerization of the LPS-caspase-11 complexes and the activation of caspase-11 non-canonical inflammasome by autoproteolysis. Activated caspase-11 non-canonical inflammasome induces GSDMD proteolysis and GSDMD pore formation, resulting in pyroptosis of the cells. Activated caspase-11 non-canonical inflammasome also induces proteolytic maturation of pro-IL-1β and pro-IL-18, and the IL-1b and IL-18 are secreted from the cells through the GSDMD pores.

**Figure 2 ijms-25-02091-f002:**
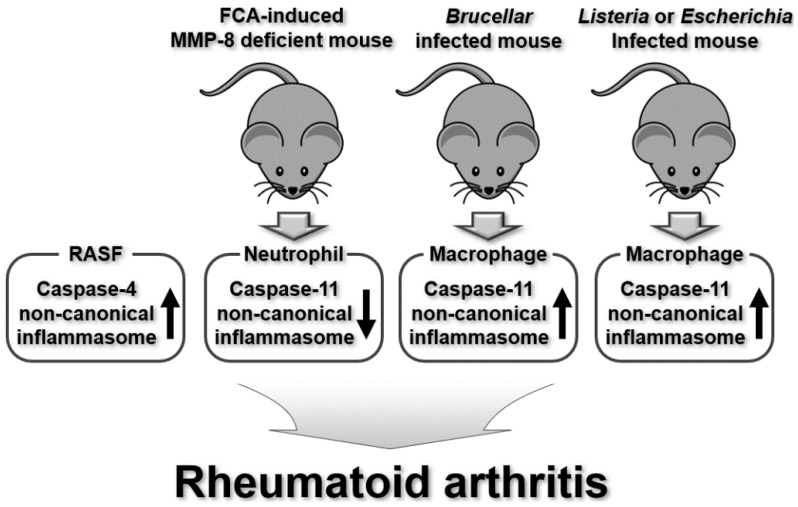
Regulatory roles of mouse caspase-11 and human caspase-4 non-canonical inflammasomes in RA pathogenesis in RASFs and RA mouse models.

**Figure 3 ijms-25-02091-f003:**
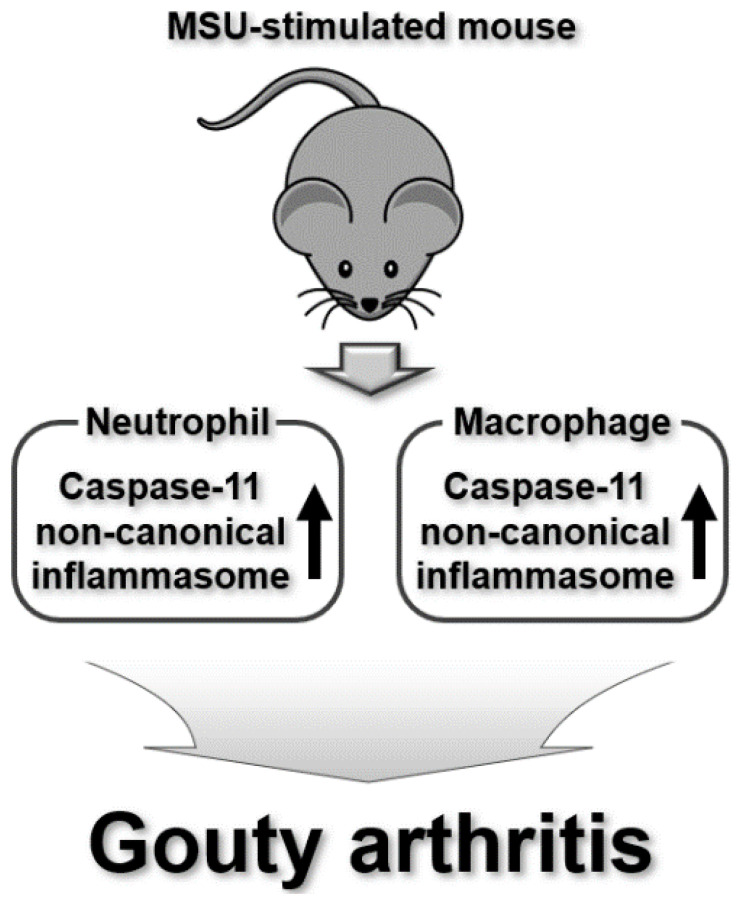
Regulatory roles of mouse caspase-11 non-canonical inflammasomes in GA pathogenesis in an MSU-induced GA mouse model.

**Figure 4 ijms-25-02091-f004:**
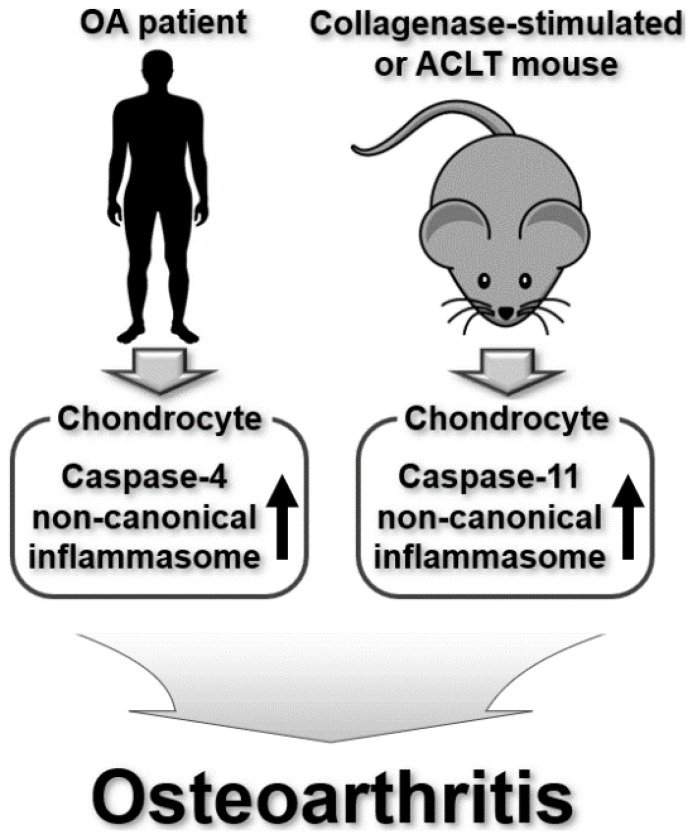
Regulatory roles of mouse caspase-11 and human caspase-4 non-canonical inflammasomes in OA pathogenesis in human OA patients and OA mouse models.

**Figure 5 ijms-25-02091-f005:**
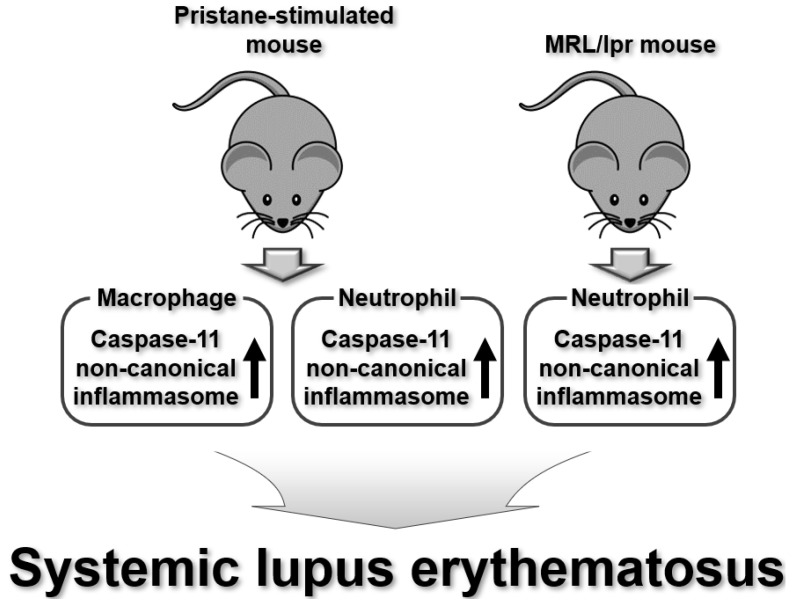
Regulatory roles of mouse caspase-11 non-canonical inflammasomes in SLE pathogenesis in lupus mouse models.

**Figure 6 ijms-25-02091-f006:**
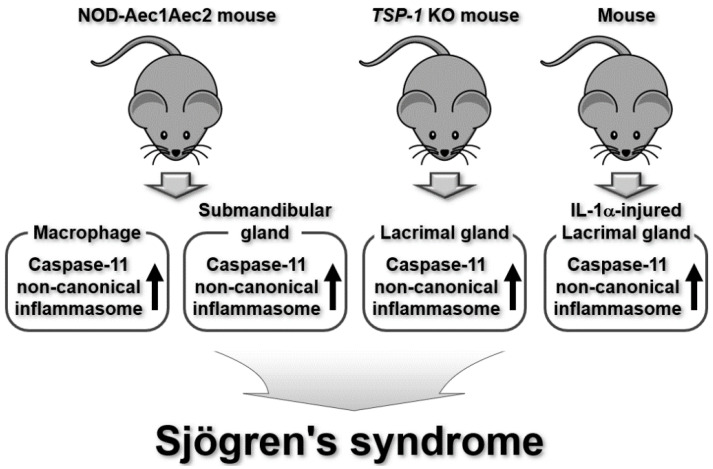
Regulatory roles of mouse caspase-11 non-canonical inflammasomes in SjS pathogenesis in SjS mouse models.

**Table 1 ijms-25-02091-t001:** Regulatory roles of non-canonical inflammasomes in rheumatic diseases.

Diseases	Caspases	Roles	Models	Ref
RA	Caspase-4	Blockade of caspase-4 non-canonical inflammasome inhibited the apoptosis of RASFs Blockade of caspase-4 non-canonical inflammasome attenuated vimentin cleavage and p53 nuclear translocation	RASFs	[56]
	Caspase-11	Neutrophil infiltration was increased in synovial tissues of MMP-8-deficient arthritic mice Symptoms of inflammatory arthritis were more severe in MMP-8-deficient arthritic mice Caspase-11 mRNA expression was downregulated in neutrophils of MMP-8-deficient arthritic mice	FCA-induced arthritic mice Neutrophils	[54]
		*Brucella* infection induced joint inflammation in mice *Brucella* infection induced caspase-11 non-canonical inflammasome activation and pro-inflammatory cytokine production in mice Caspase-11 non-canonical inflammasome-mediated pyroptotic cell death in response to *Brucella* infection was promoted in BMDMs *Brucella* infection activated caspase-11 non-canonical inflammasome, but not NLRP3 canonical inflammasome	*Brucella-*infected mice with joint arthritis BMDMs	[61]
		Circulating Prdx1 was increased in arthritic mice infected with the bacteria Blockade of Prdx1 ameliorated the symptoms in arthritic mice infected with the bacteria Prdx1 release under arthritic condition depended on caspase-11 non-canonical inflammasome activation in arthritic mice infected with the bacteria	*Listeria- or Escherichia*-infected mice with joint arthritis BMDMs	[62]
GA	Caspase-11	Absence of caspase-11 non-canonical inflammasome ameliorated GA symptoms in MSU-induced mouse model of GA Absence of caspase-11 non-canonical inflammasome reduced influx of macrophages and neutrophils to disease lesions and production of NETs in MSU-induced mouse model of GA The absence of caspase-11 non-canonical inflammasome decreased production of gout-specific pro-inflammatory cytokines in MSU-induced mouse model of GA Lower production of NETs in caspase-11-deficient neutrophils was associated with altered cofilin phosphorylation in response to MSU in mouse model of GA	MSU-induced mouse model of GA Neutrophils BMDMs	[68]
OA	Caspase-4	LPS induced apoptosis of human chondrocytes isolated from the OA patients Caspase-4 non-canonical inflammasome was activated in LPS-stimulated human chondrocytes Melatonin attenuated apoptosis and inhibited the activation of caspase-4 non-canonical inflammasome in LPS-stimulated human chondrocytes	OA patients Chondrocytes	[75]
	Caspase-11	LPS-stimulated macrophage EVs induced apoptosis of mouse chondrocytes LPS-stimulated macrophage EVs induced expression of chondrocyte catabolic and pyroptosis-associated factors in chondrocytes Disruption of caspase-11 alleviated pyroptosis and catabolic processes in macrophage EV-stimulated chondrocytes Disruption of caspase-11 ameliorated pathogenic changes in collagenase-induced and ACLT mouse models of OA	Collagenase-induced and ACLT mouse models of OA Chondrocytes	[77]
SLE	Caspase-11	Cgas-deficiency enhanced lupus symptoms and inflammatory pathology in pristane-induced lupus mice Cgas-deficiency increased caspase-11 gene expression and activated caspase-11 non-canonical inflammasome in lungs of pristane-induced lupus mice	Pristane-induced mouse model of lupus BMDMs	[87]
		GSDMD was proteolytically activated in neutrophils of SLE patients and pristane-induced and MRL/lpr mouse models of lupus GSDMD-mediated pyroptotic death of neutrophils was increased in kidneys of pristane-induced and MRL/lpr mouse models of lupus Caspase-11 non-canonical inflammasome was activated in the neutrophils of lupus mice, leading to proteolytic activation of GSDMD and extracellular DNA release through GSDMD pores Oxidized mtDNA promoted GSDMD oligomerization and GSDMD pore formation, and pyroptotic death of the neutrophils in mouse models of lupus GSDMD inhibition in neutrophils alleviated disease severity in mouse model of lupus	SLE patients Pristane-induced mouse model of lupus MRL/lpr mouse model of lupus Neutrophils	[89]
SjS	Caspase-11	Caspase-11 expression was upregulated, leading to caspase-1 activation and IL-18 production in macrophages and in submandibular glands of SjS-prone mice Apoptotic death of epithelial cells was induced in submandibular glands of SjS-prone mice	SjS-prone C57BL/6.NOD-Aec1Aec2 mouse model	[95]
		Panx1, pro-inflammatory factors, and caspase-11 were upregulated in IL-1α-injured mouse lacrimal glands Panx1 blockade increased epithelial cell progenitor engraftment into IL-1α-injured mouse lacrimal glands Inhibition of Panx1 or caspase-11 reduced inflammation in lacrimal glands of *TSP-1* KO mice by decreasing expression of pro-inflammatory factors and lymphocyte infiltration	IL-1α-injured mouse lacrimal glands Lacrimal gland of *TSP-1* KO mice	[98]

## Data Availability

Not applicable.

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
