# Peer review of "Roles of the Caspase-11 Non-Canonical Inflammasome in Rheumatic Diseases"

_ijms, 2024, doi:10.3390/ijms25042091_

Round 1
Reviewer 1 Report
Comments and Suggestions for Authors
The topic is interesting and novel, but the amount of evidence related to human rheumatic diseases is scarce.
Main evidence is confined to caspase-11 and murine disease models. It is not clear throughout the text when previous reports are referred to human disease or murine disease, which are not exactly the same and shouldn't be considered as such.
Caspase 4 (ref. 38) also processes pro-IL-1beta into IL-1beta. This mechanism is not mentioned. Does caspase 11 have the same capability?
Figure 2 does not include the relation of caspase with activation of NLRP3 as adressed in the main text.
"RA mouse" in my opinion is not correctly used, there are various "inflammatory arthritis mouse models" and to use "RA mouse" it should be defined in advance which exact model or models is referring to.
Authors tend to write excesively optimistic conclusions througout the text using terms like "key role", "demonstrate", "significantly involved", "compelling evidence", "numerous essential roles"....
The work done is remarkable and the mechanism are well described, but I would shorten many parragraphs to make it more serious, exposing the known facts in order (human/animal models), concluding with less completive or more open terms.
Conclusions do in fact adress a more "humble" view of the possible implication of non-canonical inflammasomes in rheumatic diseases pathogenesis, I believe this should be the general tone of the article.
Reviewer 2 Report
Comments and Suggestions for Authors
The review describes the importance of canonical and caspase 11/4 non-canonical inflammasome effects in rheumatic diseases.
Comments
1. The language of the paper (scientific writing) should be improved. The review is too wordy. It should be shortened.
2. Abstract: The differences between canonical and non-canonical inflammasomes should be presented here.
3. Lines 38-39: The definition of non-canonical inflammasomes should be presented here.
4. Figs1b,c repeat part of Fig 2. These Figures should be combined. The importance of Caspase 11 p26 should be defined. This should be corrected.
5. Line 151: Rheumatoid arthritis and Gout should be described in separate sections.
6. Line 153: Non-inflammatory arthritis does not exist. This should be corrected.
7. Line 158: RA definition is incorrect. This should be corrected.
8. Line 174: Which MMP is involved? This should be corrected.
9. The text on Lines 176-180 contradict to that on Lines 20-205. This should be corrected.
10. Lines 197-198; 200-222: These sentences are not clear. This should be corrected.
11. Line 264-265: The definition of OA is incorrect. This should be corrected.
12. Line 266: Spine is not involved in OA. This should be corrected.
13. Line 274: Reference is required at the end of this sentence.
14. Line 280-282: This suggestion is not evident. This should be corrected.
15. Line 288: This sentence is not clear. This should be corrected.
16. Line 409: This section should be renamed for Concluding remarks.
Comments on the Quality of English LanguageThe language of the paper (scientific writing) should be improved. The review is too wordy. It should be shortened.
Lines 197-198; 200-222: These sentences are not clear. This should be corrected.
Line 288: This sentence is not clear. This should be corrected.
Round 2
Reviewer 1 Report
Comments and Suggestions for Authors
The manuscript has been improved but still shows excesive "optimism" in terminology.
Abstract: "Interestingly, an increasing number of studies have reported essential roles for noncanonical inflammasomes in rheumatic diseases". Consider... "have reported a possible role for noncannonical inflammasomes in the pathogenesis of various rheumatic disesase mice models".
Also caspase 11 cannot be related to human diseases and is stated throughout the manuscript. Even if Caspase 11/4-5 are considered homologs, they are not the same molecule and functions may differ, specially within different species. The roles of caspase11 in mice models in my opinion cannot be directly extrapolated to human disease. Every result of caspase 11 should be related to "mice model" instead of RA, LES,...,
Figure 1: the mechanism by which non-canonical inflammasome activates NLRP3 is not clearly shown...I find it confusing.
Figure 2: Rheumatoid arthritis is a human disease, consider change the term for Inflammatory arthritis.
Round 3
Reviewer 1 Report
Comments and Suggestions for Authors
Improved.
Figure 1: as I understand NLRP3 is activated due to intracelular changes derived from GSDMD pore formation (including K+ efflux). Noncanonical inflammasome is responsible for GSDMD pores in this context but does not activate NLRP3 directly as shown in the figure. I still find confusing.
Consider redefining Inflammatory Arthritis (section 3.1, first paragraph), the definition is not completely accurate, maybe change ref. 47 for another more specific.
Figure 2: Consider a quote for RASF (RA synovial fybroblasts) for better understanding of the figure.
